# The Effect of Ni^2+^ Ions Substitution on Structural, Morphological, and Optical Properties in CoCr_2_O_4_ Matrix as Pigments in Ceramic Glazes

**DOI:** 10.3390/ma15248713

**Published:** 2022-12-07

**Authors:** Firuta Goga, Rares Adrian Bortnic, Alexandra Avram, Mioara Zagrai, Lucian Barbu Tudoran, Raluca Anca Mereu

**Affiliations:** 1Faculty of Chemistry and Chemical Engineering, Babeş-Bolyai University, 11 Arany Janos Street, 400028 Cluj-Napoca, Romania; 2Faculty of Physics, Babes-Bolyai University, 1 Kogalniceanu Street, 400084 Cluj-Napoca, Romania; 3National Institute for Research and Development of Isotopic and Molecular Technologies, 67-103 Donath Street, 400293 Cluj-Napoca, Romania; 4Electron Microscopy Center, Faculty of Biology and Geology, Babes-Bolyai University, 400006 Cluj-Napoca, Romania

**Keywords:** ceramic pigments, spinel, chromite, optical properties, coloring properties

## Abstract

The structural, morphological, and optical properties of Ni^2+^ ions substitution in CoCr_2_O_4_ matrix as ceramic pigments were investigated. The thermal decomposition of the dried gel was performed aiming to understand the mass changes during annealing. The X-ray diffraction (XRD) studies reveal a spinel-type Face–Centered Cubic structure and a secondary Cr_2_O_3_ phase when x ≤ 0.75 and a Body–Centered Tetragonal structure when x = 1. Fourier Transform Infrared Spectroscopy (FT–IR) indicated two strong absorption bands corresponding to the metal–oxygen stretching from tetrahedral and octahedral sites, characteristic of spinel structure. Ultraviolet–Visible (UV–Vis) spectra exhibited the electronic transitions of the Cr^2+^ Cr^3+^ and Ni^2+^ ions. From the UV–Vis data, the CIE color coordinates, (x, y) of the pigments were evaluated. The morphology was examined by Scanning Electron Microscopy (SEM) and Transmission Electron Microscopy (TEM) showing the agglomeration behavior of the particles. The stability, coloring properties and potential ceramic applications of studied pigments were tested by their incorporation in matte and glossy tile glazes followed by the application of obtained glazes on ceramic tiles. This study highlights the change in pigment color (from turquoise to a yellowish green) with Ni^2+^ ions substitution in the CoCr_2_O_4_ spinel matrix.

## 1. Introduction

The color of ceramic tiles, followed by the design and the technical properties, is one of the most appreciated characteristics by the customer. Thereby, there is a real need in diversifying the ceramic color palette. In addition, the use of the newer technologies for obtaining easier, faster, and qualitative colored models on tiles keeps the research in the field of ceramic pigments open. Nowadays, the demand is still on in finding new materials suitable for the challenge created by the digital decoration ink-jet printing for ceramic tiles. This application allows the use of colloidal suspensions of pigments to improve decorative features of ceramic parts and other ceramic tiles [1].

The ceramic pigments are inorganic materials which have high thermal and chemical stability. These properties allow them to be subjected to the processing conditions without losing their coloring characteristics.

Spinel compounds have a huge contribution in the field of ceramic pigments to obtain different colors like red, pink, brown, gray, or blue.

Spinels with the general formula AB_2_O_4_ represent a large family of inorganic compounds which are now used in different industrial branches as ceramic pigments [2,3,4,5], magnetic materials [6,7], catalytic materials [8], supercapacitors [9,10], due to their excellent chemical, thermal, mechanical, magnetic properties, and high melting point. Although spinels have a complex composition, they are intrinsically colored and present several advantages, when compared to the metal oxides as colorants in ceramics and enamels.

The properties of the oxide materials and consequently of spinels, are strongly correlated with the synthesis method and thermal behavior, which affects the interspersion of the components and thus defines the morphology [11]. This also influences the coloring properties of pigments.

In general, the synthesis methods employed depend especially on the final use of the material and are based on the relation between structure—morphology and properties. Consequently, there are different methods utilized in the synthesis of spinel structures, the conventional ones (co-precipitation and hydro/solvo-thermal) and non-conventional ones (sol-gel and derivatives or auto-combustion) [12,13,14]. Among them, the sol-gel synthesis offers great versatility to prepare low-cost materials with a rigorous control from the compositional and microstructural point of view. The sol-gel method is based on inorganic polymerization reactions and so, starting from molecular precursors an oxide can be easily obtained via hydroxylation-condensation reactions [1,12].

Spinel pigments can be obtained by the addition of chromophores, transition metal ions into inert matrices such as oxides or by calcination of metal-organic complexes [15]. They have the general formula AB_2_O_4_ where the A-sites are tetrahedral coordinated occupied by divalent ions as A^2+^ (like Co, Ni, Mn, Zn), and the B-sites are octahedral coordinated and occupied by trivalent ions B^3+^ (like Cr or Al) [16].

In the literature, there are studies regarding spinel oxides such as CoCr_2_O_4_, MgCr_2_O_4_, Co_(1−x)_Mg_x_Cr_2_O_4_ [17,18] and NiCr_2_O_4_ [19] which have been synthesized by different methods and are investigated for different applications due to their thermal stability, as previously noted.

Nowadays, for the blue pigment, cobalt-based materials are used [1]. There are few drawbacks for the use of cobalt, i.e., abundance, extraction, geopolitical instabilities, and the need to be used in the electronic or other industries.

In the present context, the aim of this work is to synthesize and to investigate new competitive pigments for novel technologies with as little cobalt as possible. Considering that, we studied the nickel substitution in the cobalt chromite matrix. For this investigation, an easy to use and environmentally friendly modified sol-gel method was employed for the synthesis of Co_(1−x)_Ni_x_Cr_2_O_4_ (x = 0, 0.25, 0.5, 0.75 and 1).

Structural, morphological, and optical properties were studied as a function of the nickel substitution in the spinel matrix focusing on the application as ceramic pigments.

The proposed modified sol-gel method involves metallic salts combined with the sucrose and pectin as organic precursors. The method was successfully utilized in the synthesis of other spinel oxide compounds with nanometric dimensions [20].

More than obtaining these chromite spinel pigments through an easy, cost-effective method for the first time, the aim of this work was to test their potential applicability in the ceramic tile industry. For this purpose, all synthetized pigments were embedded in two different types of glazes to also test their efficacity as ceramic pigments. The novelty of this work is to investigate the viability of all synthesized pigments to be successfully used in matte and glossy glazes applied on tiles. The coloring properties were successfully demonstrated for all samples embedded in both types of glazes.

## 2. Materials and Methods

### 2.1. Materials

The following precursors were used for the synthesis of pigments: Co(NO_3_)_2_·6H_2_O (99.5% purity, Merck, Darmstadt, Germany Merk), Ni(CH_3_COO)_2_·4H_2_O (99.5% purity, Merck, Darmstadt, Germany) and (NH_4_)_2_Cr_2_O_7_ (99.5% purity, Merck, Darmstadt, Germany). All the reagents used in this experiment are of high analytical grade and they were used as purchased, without further purification. HNO_3_ (65%, Merck, Darmstadt, Germany) was used as a pH regulator. The sucrose and pectin used to support the condensation reaction were commercial food grade.

### 2.2. Synthesis of Pigments

Concentrated precursor solutions were prepared using the metal salts (Co(NO_3_)_2_·6H_2_O, Ni(CH_3_COO)_2_·4H_2_O and (NH_4_)_2_Cr_2_O_7_) dissolved in distilled water. Separately, another solution containing the sucrose with the ratio sucrose: grams of final oxide (wt.:wt.) of 2:1 was prepared. The sol-gel process starts when mixing the precursor solutions with sucrose solution and pectin to form a gel. The addition of sucrose and pectin to the precursor solution containing the metal cations forms a polymer matrix in which the metal cations are distributed through the polymeric network structure [20]. This mechanism, and the role played by sucrose and pectin in the formation of the oxide structures, is discussed in more detail in reference [20].

The hydrolysis process occurs during the vigorous magnetic stirring (1000 RPM) of the metal solutions under strict temperature control (at 40–45 °C) and with a pH correction to around 1–1.5. After stirring, the obtained sol is left to age for 24 h at 80 °C to ensure the formation of the gel lattice, with the elimination of the water present in the pores and the final formation of a porous structure. The thermal treatment of the dried gels was performed in an electric furnace, in porcelain crucibles. The furnace temperature had an increase rate of 300 °C/h, with an isotherm plateau of 30 min, at 1000 °C.

### 2.3. Analysis Methods

The mass loss and the phase transformations occurring during the heating of the dried gel were investigated by thermal analyses (TG-DTA) employing a SDTQ600 TA Instruments (TA Instruments New Castle, DE, USA) thermal analyzer.

The thermal analysis was performed on a sample of about 10 mg placed in an alumina crucible and non-isothermally heated from 30 °C to 1000 °C at a heating rate of 10 °C/min in dynamic flow-air.

The structural characterization was carried out at room temperature by powder X-ray diffraction using a Bruker D8 Advance AXS diffractometer (Karlsruhe, Germany) with Cu Κα radiation in the 20–80° 2θ region. Crystallite size, cell arrangements and phase fractions were calculated by Rietveld refinement analysis using FullProf Suit Software (FullProff suite July–2017).

The FT-IR absorption spectra were recorded with a JASCO FTIR 6200 spectrometer in the 400–1500 cm^−1^ spectral range, with a standard resolution of ±2 cm^−1^.

Scanning Electron Microscopy (SEM, Chiyoda, Tokyo, Japan) and Transmission Electron Microscopy (TEM, Chiyoda, Tokyo, Japan) were performed employing a combined electron scanning (SE, Chiyoda, Tokyo, Japan) and transmission (TE) Hitachi HD–2700 electron microscope (Chiyoda, Tokyo, Japan) operated at a maximum acceleration voltage of 200 kV. The energy–dispersive X-ray spectrometry (EDX) was used to obtain the images with the nickel distribution in the glossy ceramic glaze.

The absorption spectra of the powder samples were obtained using a Perkin–Elmer Lambda 45 UV/Vis spectrometer (Waltham, MA, USA) with integrating sphere using the pellet technique. The powder samples mixed with BaSO_4_ were uniaxial press in a pellet matrix using a load force of 10 tons/cm^2^ to form transparent disks with diameter of 13 mm, and 2 mm thick. The spectra were recorded in the 200–850 nm wavelength range, with the wavelength accuracy ±2 nm.

The stability of pigments was checked by incorporating them in both matte and glossy tile glaze, using a 2% pigment addition. Thermal treatment for glaze melting was performed at 1200 °C for 6 h at maximum temperature. The glazes were characterized using a color spectrophotometer spectro–guide series (BYK Gardner, Los Angeles, CA, USA).

## 3. Results

### 3.1. Thermal Analysis

The thermal decomposition of dried precursor gel was investigated to understand the details of the decomposition process, but most importantly, to determine the temperature at which nucleation and crystallization takes place. This is imperative from a technological point of view and the potential application of the studied pigments in the industry. The thermogravimetric (TG) and differential thermal analysis (DTA) thermograms (20–1000 °C) of the CoCr_2_O_4_ dried gel are presented in Figure 1.

The decomposition of the dried gel follows two steps:the removal of the residual water (the absorbed and the coordinated water) with a mass loss of about 5.09% (30–61 °C),the weight loss of the organic fragment with formation of volatile compounds, with a total mass loss of approximately 9.13% (61–1000 °C).

A significant mass loss takes place up to 400 °C, after which no significant losses are recorded. Overall, the total mass loss at 1000 °C was 15.02% and is assigned to the drying process of the gel, in which much of the organic precursor has been removed. Prior to 900 °C, a small mass loss of 1.03% is attributed to the volatile compounds formed and remained inside the material pores. The DTA curve presents a big exothermic peak which unfolds in the 20–800 °C range confirm the two listed decomposition steps. The peak corresponding to the elimination of the organic fragments can be described by the following temperatures: T_onset_ = 318.35 °C, T_peak_ = 331.02 °C and T_end_ = 352.76 °C.

The dried gels were subjected to thermal treatment at 1000 °C for 30 min to ensure the compound crystallization.

### 3.2. Structural Characterization of Spinels

Figure 2 presents the XRD patterns of the Ni^2+^ ions substitution in CoCr_2_O_4_ spinel matrix after the thermal treatment in air atmosphere at 1000 °C for 30 min.

Both NiCr_2_O_4_ and CoCr_2_O_4_ oxides crystallize into normal spinel structures in the AB_2_O_4_ form with Co^2+^ and Ni^2+^ ions occupying the tetrahedral A sites and Cr^3+^ ions occupying octahedral B sites.

The XRD patterns present only intense and sharp peaks characteristic to the crystalline phases.

The Co_(1−x)_Ni_x_Cr_2_O_4_ spinel (when x ≠ 1) crystallizes in Face-Centered Cubic structure and is indexable to the space group Fd-3m (no. 227). This is consistent with the standard values of Face-Centered Cubic phase (00-022-1084 PDF files). On the other hand, NiCr_2_O_4_ crystalizes in the Body-Centered Tetragonal structure, the space group I41/amd (no. 141). So, a transition from the cubic spinel (when x < 1) to a Body Centered Tetragonal structure (when x = 1) is observed.

For samples with x < 1 the XRD data analysis indicated the formation of two crystalline phases: a well crystallized phase Co_(1−x)_Ni_x_Cr_2_O_4_ spinel with similar structure to CoCr_2_O_4_ (00-022-1084 PDF files) as the major phase, and Cr_2_O_3_ (00-038-1479 PDF files) as a secondary phase. Crystallinity dependent properties of crystals are attributed to their crystallite size. Larger crystallites develop sharper peaks on the XRD pattern for each crystal plane. The width of a peak is related to its crystallite size [21]. The formation of Cr_2_O_3_ impurity phase is common during the synthesis process of NiCr_2_O_4_ and is often reported in the literature [22,23].

The XRD peaks at 2θ values observed in the NiCr_2_O_4_, pattern, match the (112), (200), (103), (211), (202), (004), (220), (312), (106), (321), (224) and (400) crystalline planes of the Body-Centered Tetragonal spinel structure (01-088-0109 PDF files). No other secondary peaks were detected, suggesting that the Ni^2+^ substitution occurred.

The Rietveld refinement fittings were carried out to investigate the changes in the crystal structure by using FullProf software. The experimental recorded data points are represented with black circles, the structural model fit is represented with a red solid line, the structural model fit is represented by the blue solid line and the vertical green bars represent the Bragg positions. Up until the best possible convergence, the refinement was continued. The good fit is evidenced in Figure 3 where the small variance in the difference curve can be observed.

The average crystallite sizes, the lattice parameters and the average strain and standard deviation calculated with the FullProf software are listed in Table 1 and Table 2. Thereby, the average crystallite sizes increase with the increase in concentration of Ni^2+^ ions in Co_(1−x)_Ni_x_Cr_2_O_4_ spinel starting from 39.9 nm when x = 0 and reaching 99.42 nm when x = 0.75. The average crystallite size of NiCr_2_O_4_ (when x = 1) was 58.98 nm.

The evolution of the average crystallite size, cell parameter, density, and unit cell volume as function of nickel concentration are represented in Figure 4. Thereby, the lattice constant value of Co_(1−x)_Ni_x_Cr_2_O_4_ spinel when x ≤ 0.75 structure was measured to be 8.33 Å, and it is in concordance with the literature data [24]. This large value of the lattice constant was assigned to the disordering of cations in the spinel structure of CoCr_2_O_4_ and due to the exchange of tetrahedral A-site Co^2+^ ions with octahedral B-site Cr^3+^ ions [2].

When the Ni^2+^ content is increased, the lattice parameter and volume unit decrease (Figure 4a,b). This decrease is assigned to the difference of ionic radius of Ni^2+^ and Co^2+^, i.e., with nickel having a smaller radius. Similarly, the density increases with the increase of nickel concentration (Figure 4d).

Considering the targeted application as pigments, the crystallite size plays a determining role in the coloration capacity as being conditioned/size dependent on the specific surface.

### 3.3. FT–IR Spectroscopy

The FT–IR spectra of Co_(1−x)_Ni_x_Cr_2_O_4_ (x = 0, 0.25, 0.5, 0.75 and 1) samples are presented in Figure 5. Two strong absorption bands centered at 516, 620 cm^−1^ with a shoulder at 666 cm^−1^ are observed. These absorption bands are the spinel characteristic peaks and depend on the vibration of the cations at the B site [25,26,27].

As can be observed in FTIR spectra, the peak centered at 525 cm^−1^ from the sample with x = 0 shifts to a lower frequency, at 504 cm^−1^, with the increasing nickel content x = 1 (in NiCr_2_O_4_). The crystal theory of the field stabilization energy advises that the Cr^3+^ and Co^2+^ are incorporated into the octahedral and tetrahedral interstices site and compared to the Cr^3+^ and Ni^2+^, which by occupying the octahedral interstices site own a smaller ligand gravity and splitting energy of d orbital due to the smaller amount of electric charge will cause the peak to shift to lower frequencies [25,26,28]. The second peak at 620 cm^−1^ also shifts towards lower frequencies with the increase of nickel content and has the same origin as the peak from 516 cm^−1^ [25]. Minor changes in the observed band shape can be assigned to the particle size, a similar behavior being observed in CoCr_2_O_4_ samples presented in the literature [29].

### 3.4. Morphological Characteristics

SEM and TEM micrographs of Co_(1−x)_Ni_x_Cr_2_O_4_ samples, with x = 0, 0.25, 0.5, 0.75 and 1, are presented in Figure 6. All samples have the tendency to agglomerate their small particles generating irregular and larger aggregates with various rhombohedral-like shapes. Additionally, no significant differences in the spinel microstructure with the increase of Ni^2+^ ion content was observed, suggesting a similar morphology for all compositions. The estimated crystallite sizes, according to TEM analysis, confirm the calculated values from XRD.

### 3.5. UV-Vis Spectroscopy

The Ni^2+^ substitution of Co^2+^ in CoCr_2_O_4_ matrix was investigated by UV-VIS spectroscopy. The optical absorption spectra of the Co_(1−x)_Ni_x_Cr_2_O_4_ (x = 0, 0.25, 0.50, 0.75 and 1) recorded in the wavelength range from 250–850 nm are presented in Figure 7. The recorded spectra of the synthesized nanoparticles are very similar to the spectra of CoCr_2_O_4_ [29] and, also of NiCr_2_O_4_ [29] as reported in the literature. The band around 259 nm can be attributed to the charge-transfer transitions between O^2−^ and Cr^3+^ [17,29].

The incorporation of nickel ions in the CoCr_2_O_4_ matrix leads to the broadening of the band centered around 350 nm, band assigned to a charge transfer of Ni^2+^ cation [25,30] and to a decrease in intensity of the bands in the 574–700 nm wavelength range. Additionally, the absorption bands of Co_(1−x)_Ni_x_Cr_2_O_4_ samples centered at 574 nm, 612 nm, and 660 nm shifts with 10 nm to higher wavelengths and the band centered at 760 nm shifts to lower wavelengths with the increase of Ni^2+^ content.

The shoulder present in all the spectra at around 460 nm is attributed to the intrinsic d-d transition of Cr^3+^ ions [31,32,33]. A decrease in the intensity of the absorption bands at wavelengths higher than 500 nm was observed with the increase of Ni^2+^ ions in the matrix. The absorption bands at about 574 nm, 612 nm are frequently observed in cobalt chromite oxides and can be attributed to the spin-allowed electronic transition ^4^A_2_(F)→^4^T_1_(P) of Co^+2^ ions in A site. The absorption band from 612 nm it is also assigned to the transition ^4^A_2g_→^4^T_2g_ of Cr^3+^ ion at the B site [17,25]. The band from 660 nm and 750 nm corresponds to ^4^A_2_→^4^T_2_ and ^3^A_2_→^3^T_1_ transitions and are due to the d-d transition of Ni^2+^ ions in the tetrahedral coordinated O^2−^ environment [20,32]. The bands from about 750 nm and 820 nm become more visible with the increase of Ni^2+^ ions in the matrix and confirm the incorporation of Ni^2+^ in tetrahedral coordination [34].

### 3.6. CIE Diagram of the Obtained Pigments

The UV-Vis data were used to determinate the CIR color coordinates, (x, y) of the pigment samples. The obtained color parameters are listed in Table 3 and represented in the CIE diagram illustrated in Figure 8. As can observed, the color coordinates slightly vary from (0.3865, 0.3295) when x = 0.25 to (0.3118, 0.2762) when x = 1 and ranging in between for x = 0.5 and 0.75. Increasing the nickel concentration in the matrix leads to a shift from a pink-turquoise to a greenish-blue color. This observation is correlated with the literature data [10] and the UV-Vis absorption spectra described previously.

The pigments were embedded in both matte and glossy tile glazes to test their applications in the ceramic tile industry. Thereby, the homogeneous glasses were prepared by incorporating spinel nanopowders into ceramic glazes which were further applied on ceramic substrates and were finally subjected to thermal treatments.

Figure 9 presents a more comprehensive view of the pigments before and after their embedding. As can be observed, the pigment powders exhibit bluish tones that tend to become greener with the progression of Ni^2+^ substitution up to a yellowish green with the full Co^2+^ replacement.

The color of pigment powders and those embedded in glazes, in terms of color parameters (L* = lightness, a* = red-green axe values, b* = yellow-blue axe values, G* = gloss) are presented in Table 4. A major difference that can be observed is that of the ‘L’ parameter, that ranges between 31.09 and 33.49 for the matte glaze and from 40.69 to 45.72 for the glossy one. The gloss parameter (‘G’) is lower for matte glazes (50.5–58.1) and higher for glossy ones (83.8–86.8), due to the nature of the glaze itself. This can be explained by the fact that the latter reflects light due to its smoother surface, whereas the former tends to scatter it. Therefore, pigments embedded in glossy glazes will naturally become ‘brighter’ in nature.

The a* and b* coordinates of the glazes increase with increasing nickel substitution in pigment which impacts the color by his chromophore capacity and also by the impact on particle size. Generally, this ranging can also be correlated with different synthesis methods, with different factors such as synthesis conditions (temperature, stoichiometry, and pH) or particle size. For example, the growth in b* value coordinate was correlated with the increase in particle size in CoAl_2_O_4_ for blue pigment [1]. In glazes, b* values are highly reduced to a range of −6.7 and −0.4, also with greener hues [1,35].

An increase in a* (red-green) and b* (yellow-blue) values can also be observed with an increase in Ni substitution. Higher a* values tend to lead to yellowish-brown hues and higher b* values to reddish-brown ones. This can be better observed by the color progression in Figure 9. Once embedded in the glazes, the color of the pigment becomes darker and more intense due to thermal treatment.

The experimental results indicate that the synthesized Co_(1−x)_Ni_x_Cr_2_O_4_ spinel (0.25 ≤ x ≤ 1) oxide nanopowders can be successfully used as ceramic pigments due to their coloration capacity and thermal resistance. The thermal and chemical stability of the Co_(1−x)_Ni_x_Cr_2_O_4_ spinel pigment in the glazes is high enough to obtain a uniform color distribution of the product and is not affected by melting in the firing process.

### 3.7. EDX Mapping

The nickel substitution in the spinel was evidenced after embedding in glossy ceramic tile by using the Energy-Dispersive X-ray Spectrometry (EDX) from SEM. Figure 10 presents the EDS element layered image including Ni Kα1 of the ceramic glossy tile cross-section as a function of nickel substitution in the spinel. A uniform distribution of the nickel in the cross-section of the glaze is observed for all samples and the partial and total substitution of nickel is evidenced in Figure 10.

## 4. Conclusions

The paper reports structural, morphological, and optical properties of new ceramic pigments-based spinel structure, obtained by sol-gel route. The structural characterization is consistent with the substitution of chromophores Co^2+^ ions with the Ni^2+^ ions in the CoCr_2_O_4_ matrix. The XRD analysis revealed the formation of Co_(1−x)_Ni_x_Cr_2_O_4_ spinel-type Face-Centered Cubic structure as a principal phase and Cr_2_O_3_ as a secondary one when x ≤ 0.75. A decrease in unit cell parameter and the unit cell volume was achieved with an increase of x. When x = 1, NiCr_2_O_4_ crystallizes in the Body-Centered Tetragonal spinel as a single phase when compared to lower ‘x’ values when a secondary phase appears. The increase of nickel substitution in the matrix has a pronounced increase in the size of the crystallites from 39.9 nm (for x = 0) to 99.42 nm (for x = 0.75). FT-IR spectra confirmed the spinel structure formation and the elimination of all the organic fragments. The UV-Vis absorption spectra presented the bands corresponding nickel ions located at the A site and the chromium ions located at the B sites. Adjusting the nickel content in the CoCr_2_O_4_ matrix the color of the pigment can be easily controlled, as it can also be seen in the CIE diagram chromaticity. SEM and TEM microscopy confirmed evidence of the powder morphology and the tendency of nanoparticles to agglomerate. The CIELab coordinates of the pigments embedded in glossy and matte tile glazes reveal the color ranging of the two glazes. Additionally, the elemental EDX distribution of Ni Kα1 confirms the homogeneous and uniform distribution of the pigment in the glossy glaze after firing. The obtained pigments can be successfully applied in glaze tiles and ceramics.

## Figures and Tables

**Figure 1 materials-15-08713-f001:**
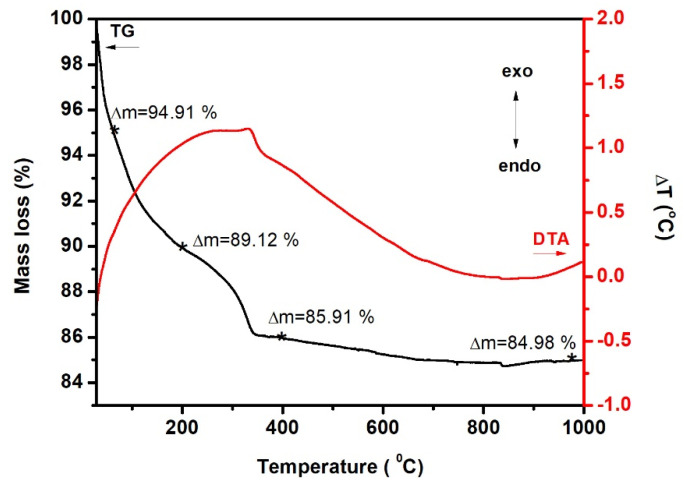
The TG and DTA curves for CoCr_2_O_4_ dried gel.

**Figure 2 materials-15-08713-f002:**
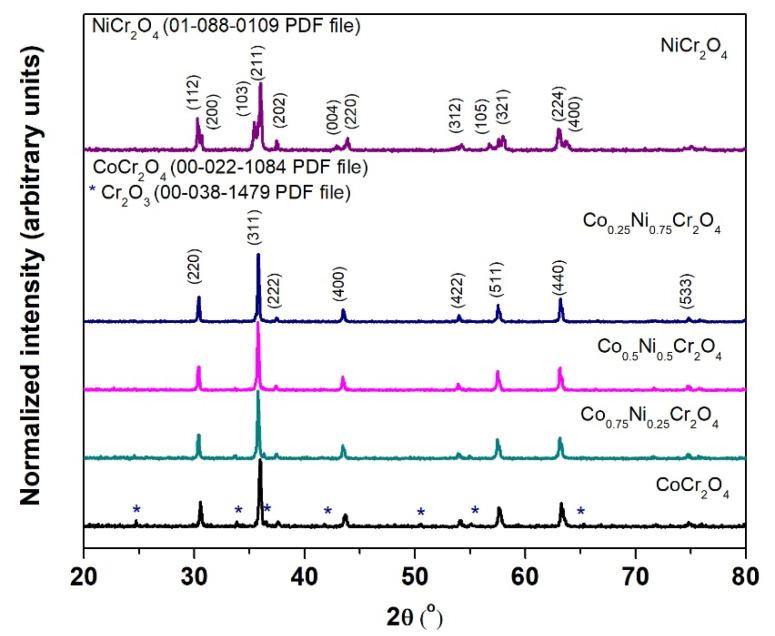
XRD patterns for Co_(1−x)_Ni_x_Cr_2_O_4_ pigments annealed at 1000 °C in air.

**Figure 3 materials-15-08713-f003:**
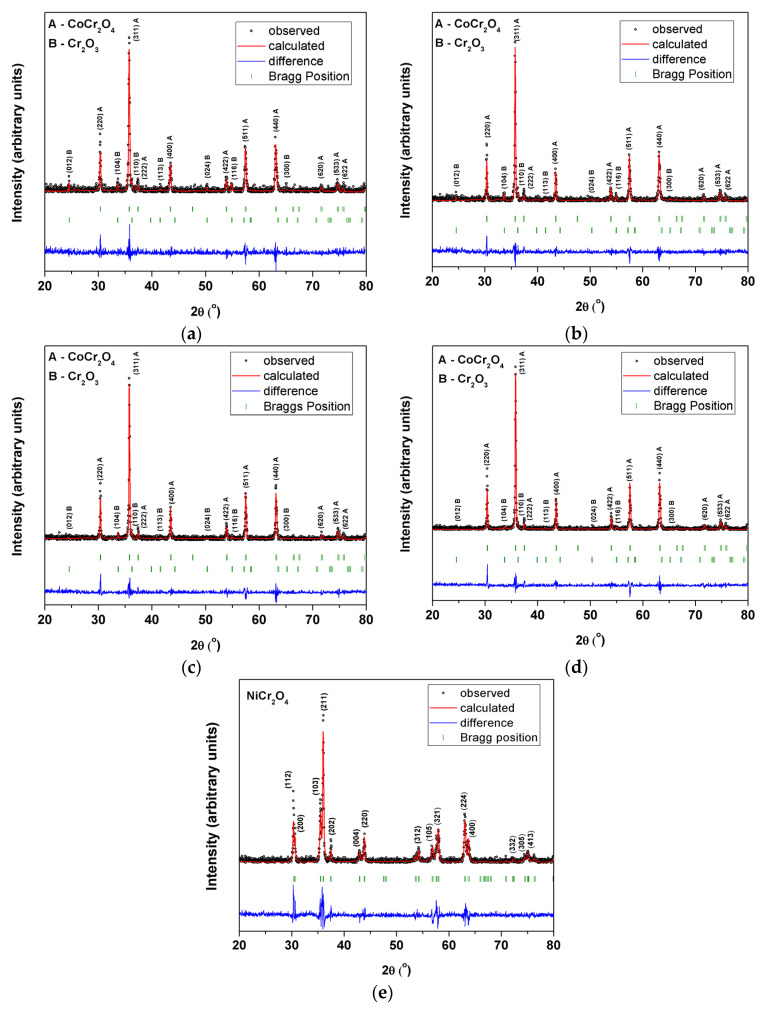
Rietveld analysis of powder-diffraction pattern of (**a**) CoCr_2_O_4_ (**b**) Co_0.75_Ni_0.25_Cr_2_O_4_ (**c**) Co_0.5_Ni_0.5_Cr_2_O_4_ (**d**) Co_0.25_Ni_0.75_Cr_2_O_4_ (**e**) NiCr_2_O_4_.

**Figure 4 materials-15-08713-f004:**
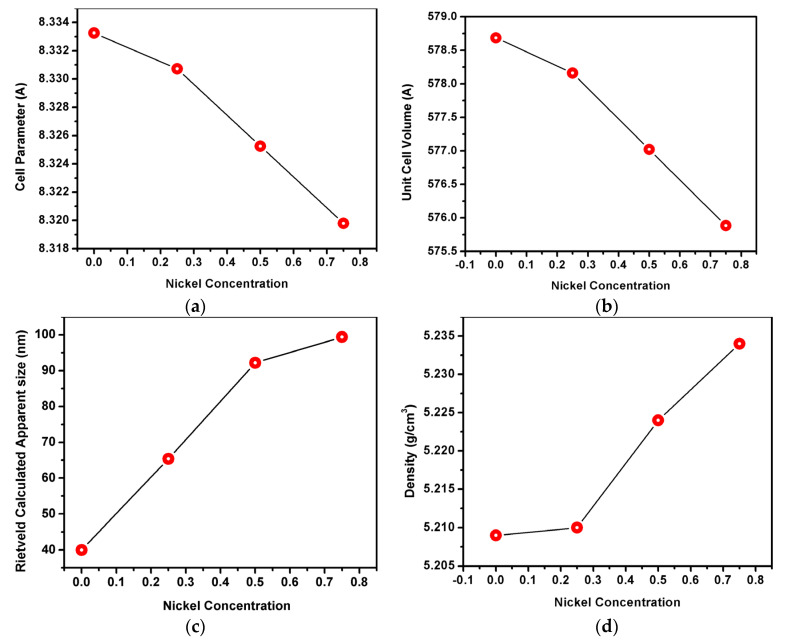
Variation of (**a**) cell parameter, (**b**) unit cell volume with the nickel concentration, (**c**) Rietveld Calculated Apparent size and density (**d**).

**Figure 5 materials-15-08713-f005:**
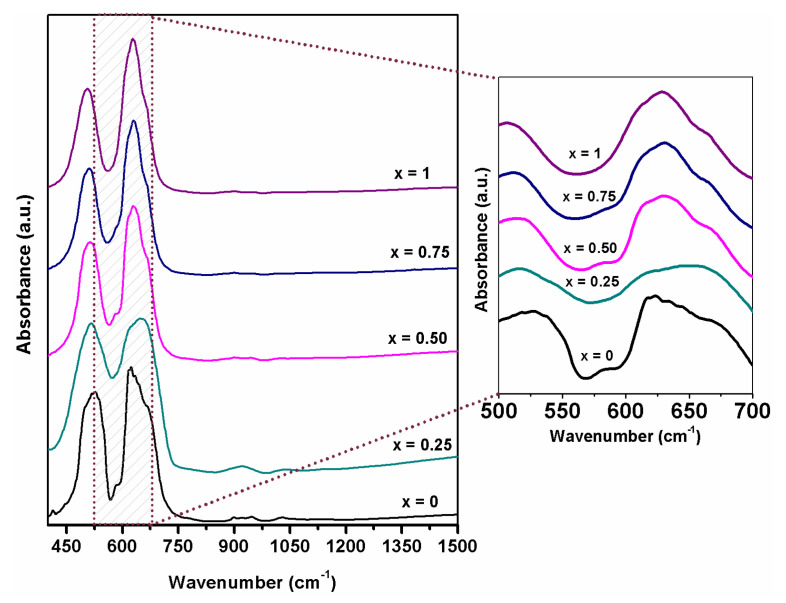
FT-IR spectra of Co_(1−x)_Ni_x_Cr_2_O_4_ (x = 0, 0.25, 0.5, 0.75 and 1).

**Figure 6 materials-15-08713-f006:**
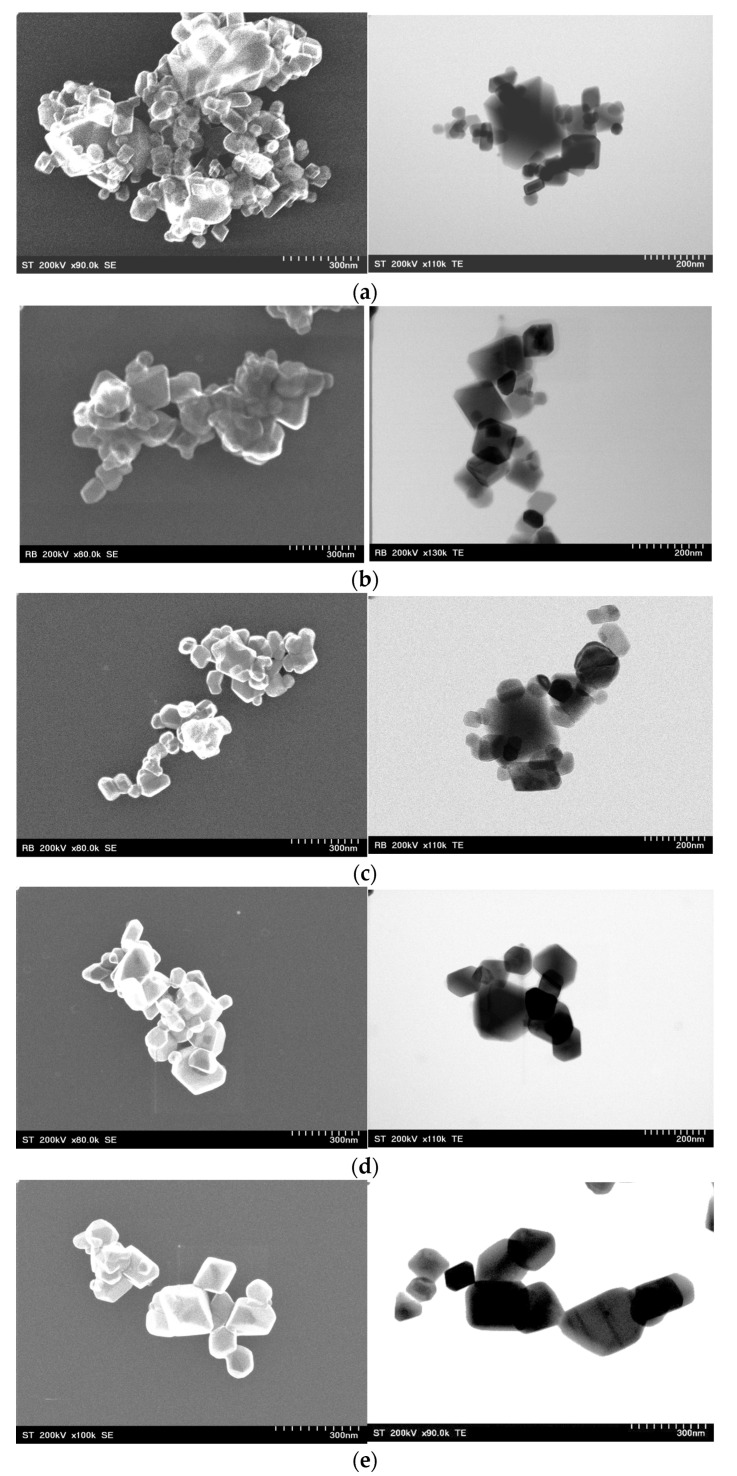
SEM and TEM images of (**a**) CoCr_2_O_4_ (**b**) Co_0.75_Ni_0.25_Cr_2_O_4_ (**c**) Co_0.5_Ni_0.5_Cr_2_O_4_ (**d**) Co_0.25_Ni_0.75_Cr_2_O_4_ (**e**) NiCr_2_O_4_.

**Figure 7 materials-15-08713-f007:**
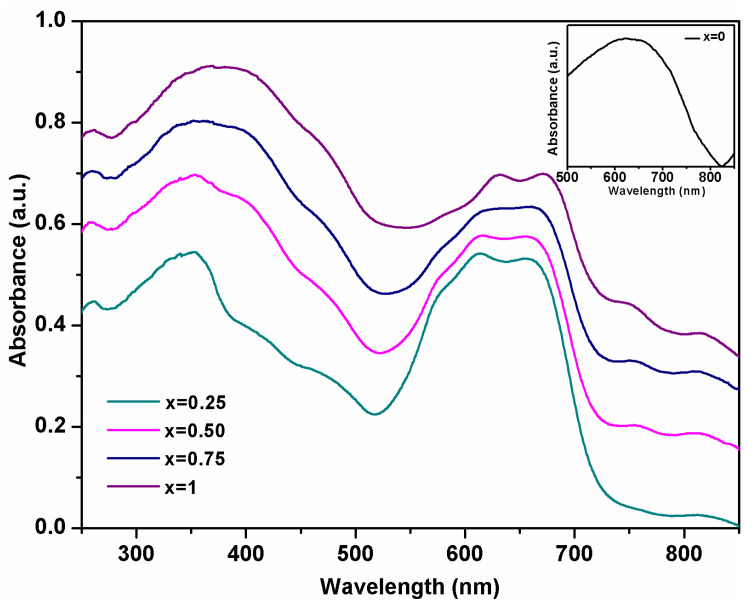
UV–Vis spectra of Co_(1−x)_Ni_x_Cr_2_O_4_ pigment (x = 0, 0.25, 0.5, 0.75 and 1).

**Figure 8 materials-15-08713-f008:**
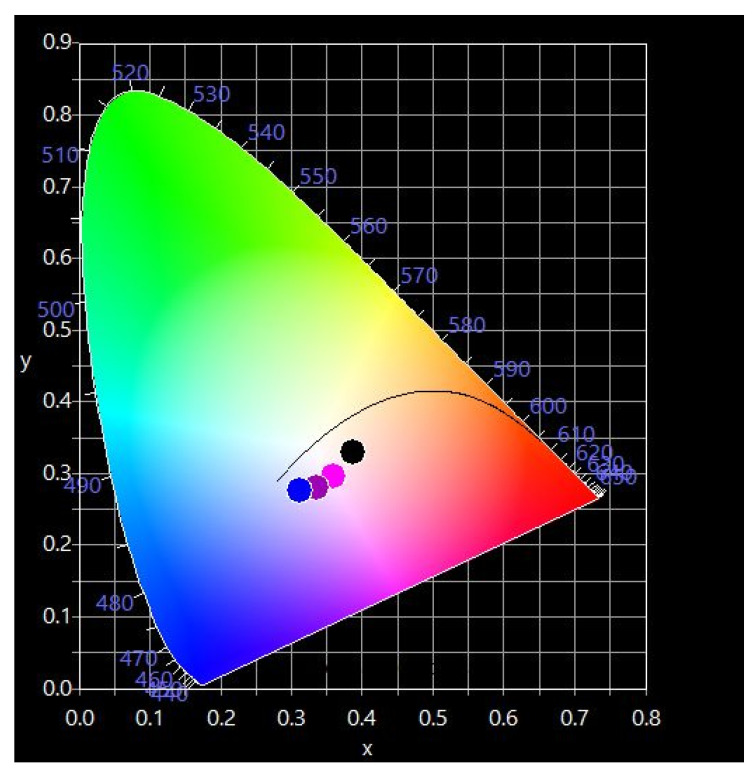
Chromaticity diagram. The place of the pigment nanopowders.

**Figure 9 materials-15-08713-f009:**
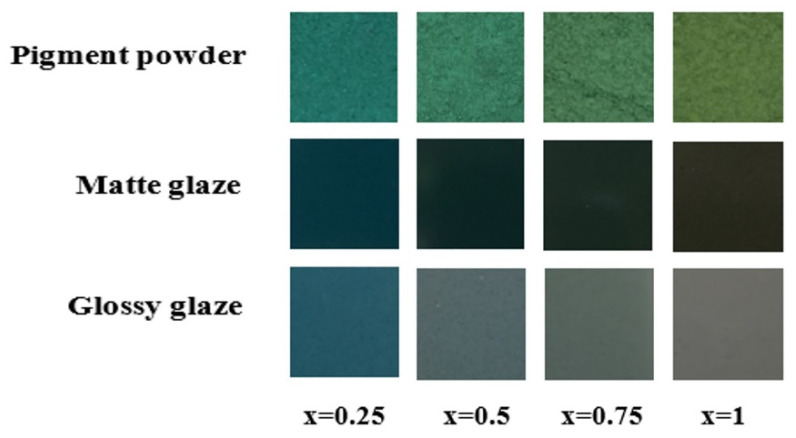
The Co_(1−x)_Ni_x_Cr_2_O_4_ pigment Nano powders before after their embedding in matte and glossy glaze.

**Figure 10 materials-15-08713-f010:**
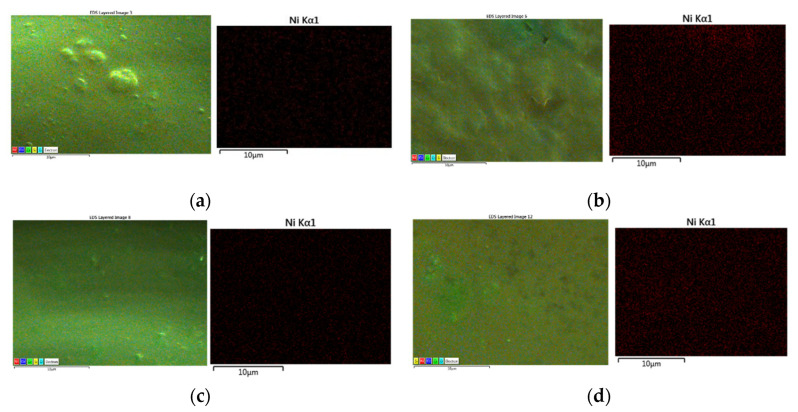
EDX element layered image including Ni Kα1 of the ceramic glossy tile cross-section as a function of nickel substitution in the chromite spinel (Co_(1−x)_Ni_x_Cr_2_O_4_) where (**a**) x = 0.25, (**b**) x = 0.5, (**c**) x = 0.75 and (**d**) x = 1.

**Table 1 materials-15-08713-t001:** Calculated crystallographic parameters.

Ni Concentration (%)	Average Apparent Size and Standard Deviation (nm)	Cell Parameter a (Å)	Cell Parameter c (Å)	Average Maximum Strain and Standard Deviation
0	39.902 (9.847)	8.33325		1.9764 (0.0007)
0.25	65.478 (26.594)	8.33072		1.9764 (0.0007)
0.5	92.199 (52.736)	8.32525		1.9520 (0.0005)
0.75	99.428 (61.333)	8.31979		1.9764 (0.0007)
1	58.983 (13.616)	5.84012	8.42349	5.3902 (0.0046)

**Table 2 materials-15-08713-t002:** Calculated phase distribution.

Ni Concentration (%)	Mass (%) Phase 1	Mass (%) Phase 2 (Cr_2_O_3_)	Density Phase 1 (g/cm^3^)	Density Phase 2 (g/cm^3^)
0	90.91 (2.25)	9.09 (0.85)	5.209	5.229
0.25	93.22 (2.23)	6.78 (0.74)	5.209	5.210
0.5	97.12 (2.32)	2.88 (0.50)	5.224	5.248
0.75	95.36 (2.00)	4.65 (0.71)	5.234	5.250
1	-	-	5.241	-

**Table 3 materials-15-08713-t003:** Calculated values of the pigment nanopowder color.

Sample Co_(1−x)_Ni_x_Cr_2_O_4_ (% Ni)	Color Coordinates (x, y)		
	x	y	Representation in the CIE Diagram
x = 0.25	0.3865	0.3295	dark circle
x = 0.50	0.3585	0.2957	pink
x = 0.75	0.3361	0.2801	purple
x = 1	0.3118	0.2762	blue

**Table 4 materials-15-08713-t004:** The CIELab coordinates of pigments embedded in opaque tile glazes.

Pigment			Matte Glaze			Glossy Glaze		
CIELab Coordonates	L*	a*	b*	G	L*	a*	b*	G
Co_0.75_Ni_0.25_Cr_2_O_4_	32.32	−9.78	−3.2	50.5	40.69	−10.63	5.65	86.1
Co_0.5_Ni_0.5_Cr_2_O_4_	31.31	−6.67	1.2	57.6	42.03	−9.45	0.15	83.8
Co_0.25_Ni_0.75_Cr_2_O_4_	31.09	−4.07	3.82	58.1	43.34	−6.63	4.98	86.7
NiCr_2_O_4_	31.71	−0.62	6.74	52.7	45.72	−1.63	11.10	86.8

## Data Availability

Not applicable.

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
