# Peer review of "The Effect of Ni2+ Ions Substitution on Structural, Morphological, and Optical Properties in CoCr2O4 Matrix as Pigments in Ceramic Glazes"

_materials, 2022, doi:10.3390/ma15248713_

Round 1

Reviewer 1 Report

The submitted paper deals with the preparation and characterization of Co(1-x)NixCr2O4 pigments to be used in ceramic glazes.

The synthesis was performed by using four loads of Ni (from 0.25 to 1); several analytical techniques have been applied to comprehensively analyze the obtained pigments. The effects of the substitution of Co with Ni have been examined from the chemical, crystallographic, morphological, and colorimetric point of view. The distribution of Ni from the pigment in tile glazes and their color features have also been discussed.

The paper is well organized. The experimental plan is properly structured and the results have been adequately discussed. In my opinion, only minor revision is needed.

The following are my comments and suggestions:

- Lines 24-25: please, rephrase the sentence.

- Lines 43, 213, 239, and 295: cite the references in numeric order.

- Lines 64-66 (Structural…employed.): revise the phrase, its meaning is not clear; then, move it at the end of the paragraph.

- Lines 132-135: specify the ranges of temperature at which the reported weight losses were detected.

- Line 174: replace “Figure 2” with “Figure 3”.

- Figure 2 and Figure 3: the graphs are difficult to read, increase the font size.

- Section 3.4: why two color spaces (xy and Lab) have been used for the color characterization of the pigments?

- Table 3: replace "Cielab" with "CIELab".

- Figure 10: regions of the same size have to be compared; on the element layered image, the bar scales and the legends are not readable; in the maps, contrast and/or brightness should be tuned to improve visibility of the Ni distribution; on the caption text, specify the Ni load for each image.

- Line 335 and lines 179-181: report the same values; I would suggest using the exact results rather than the approximate values, on lines 179-181 at least.

Author Response

Respectfully, authors thank reviewer no. 1 for the evaluation of our research work, the appreciation of our research results and recommendation for publication in Materials after some minor revision.

The authors answered to all the reviewer’s observation. We also adjust Figures 2 to 7 and 10 adding additional information and adjusting the clarity. Please se below and attached the authors comments.

Lines 24-25: please, rephrase the sentence.

The authors reformulated the phrase.

Lines 43, 213, 239, and 295: cite the references in numeric order.

The authors realized the error and change the references to be in numeric order.

Lines 64-66 (Structural…employed.): revise the phrase, its meaning is not clear; then, move it at the end of the paragraph.

The phrase was restructured to: “Structural, morphological, and electronic properties were studied as a function of the nickel substitution in the spinel matrix focusing on the application as ceramic pigments.” and improvements were added to the introduction section.

Lines 132-135: specify the ranges of temperature at which the reported weight losses were detected.

The discussion regarding the ranges of temperatures and weight loss was improved upon.

Line 174: replace “Figure 2” with “Figure 3”.

The Figure entitled “Rietveld analysis of powder-diffraction pattern” was corrected numbered to Figure 3.

Figure 2 and Figure 3: the graphs are difficult to read, increase the font size.

The font size of Figure 2 and 3 was increased accordingly to the fonts of the other figures.

Section 3.4: why two color spaces (xy and Lab) have been used for the color characterization of the pigments?

CIELab coordinates were used to characterize the powders and the glazes but the authors consider that an extra value can be added to the study with the introduction of the CIR chromaticity color coordinates, (x, y) and the CIE 1931 chromaticity diagram generated from the UV-Vis spectra of the pigments.

Table 3: replace "Cielab" with "CIELab".

We corrected this accordingly.

Figure 10: regions of the same size have to be compared; on the element layered image, the bar scales and the legends are not readable; in the maps, contrast and/or brightness should be tuned to improve visibility of the Ni distribution; on the caption text, specify the Ni load for each image.

The authors apologies for choosing images with scale discrepancies and added new images to keep the same scale. Also, the nickel substitution amount was mentioned in the figure legend.

Line 335 and lines 179-181: report the same values; I would suggest using the exact results rather than the approximate values, on lines 179-181 at least.

The exact values were provided in the lines specified by the reviewer.

Respectfully, we thank the Editor for his thoughtful cooperation, and we believe our R1 manuscript can meet the journal’s high standards and will be acceptable for publication.

On behalf of authors, 

Raluca Anca Mereu

Reviewer 2 Report

I have studied the manuscript entitled "Influence of Co2+ substitution with Ni2+ on structural, morphological and optical properties of Co(1-x)NixCr2O4 and its application as a pigment in ceramic glazes" and provided some comments. Before further consideration in the journal of "Materials", the following comments need to be addressed:

1. In my opinion, the title must be improved or changed. It's confusing.

2. In the whole manuscript and especially the abstract, please define acronyms before using them in sentences. The authors must write the full expressions first instead of directly using the abbreviations. For example, scanning electron microscopy (SEM). If the abbreviation is used only once in the abstract, you must write its full name.

3. The abstract must be improved by including the optical and FTIR discussions.

4. The introduction is not clear enough and it is difficult for the reader to find a relation between ideas with the text in general. Some parts of the current introduction lack consistency and are not easy to follow and should be rewritten. Furthermore, the authors should emphasize their novelties concerning others in this field. They must underline what is the motivation to start to study exactly this topic. Besides they should also mention explicitly, what the purpose of this work is and what they aim at. Besides, it does not provide an appropriate context for reviewing the literature and does not properly answer the question 'Why'.

5. In the manuscript, please avoid using "etc.".

6. In the experimental section please give more explanation about how the synthesized samples have been measured for their optical properties.

7. The manuscript contains relatively a high proportion of self-citations.

8. Please provide a larger view of the FTIR spectra in the range of 500-700 cm-1 for all samples for comparison.

9. I recommend the authors discuss deeply the obtained results by linking them together to get a scientific finding.

10. Rietveld analysis plots must be "Figure 3" instead of "Figure 2".

11. In Figure 2. (especially XRD patterns for NiCr2O4), some main and intense peaks seem to be split into two peaks. please explain about it.

12. The quality of Figure 4 and Table 1 are not acceptable and suitable. Table 1 is not well organized.

13. In Figures 2 and 3 the authors talk about 5 samples despite other Figures. It's kind of confusing for readers. Moreover, in the whole manuscript some sentences are vague, please express them vividly and adjust all of them to a unique state (from the point of view of the number of samples). For instance:

"varying the quantity of Ni2+ ions in the CoCr2O4 spinel matrix"

"XRD patterns of different Co2+ substitution with Ni2+ ions in CoCr2O4 spinel matrix"

"The influence of Co2+ ions substitution with Ni2+ ions on the structural, morphological and optical properties of Co(1-x)NixCr2O4 (x=0.25, 0.5, 0.75 and 1) was investigated"

14. The conclusions of the work don’t convince me and it should be rewritten. The conclusion must be quantitative and comparative. The authors should clarify their findings in this section. Please state the most valuable sample among the other samples that gave the highest/good performance. Besides, the conclusion must be in the past tense.

15. Finally, there are some grammatical, typos, and technical errors. besides a few unclear sentence structures that exist in the whole manuscript. Please proofread them before more consideration. 

Author Response

Respectfully, authors thank reviewer no. 2 for the evaluation of our research work, the appreciation of our research results and recommendation for publication in Materials.

The authors answered to all the reviewer’s observation. We also adjust Figures 2 to 7 and 10 adding additional information and adjusting the clarity. Please see below and attached the authors comments:

1. In my opinion, the title must be improved or changed. It's confusing.

At the reviewer’s suggestion the titles has been changed to be more clear:

The effect of Ni2+ ions substitution on structural, morphological and optical properties in CoCr2O4 matrix as pigments in ceramic glazes”

2. In the whole manuscript and especially the abstract, please define acronyms before using them in sentences. The authors must write the full expressions first instead of directly using the abbreviations. For example, scanning electron microscopy (SEM). If the abbreviation is used only once in the abstract, you must write its full name.

The abstract was improved by adding all acronyms, as reviewer suggested

3. The abstract must be improved by including the optical and FTIR discussions.

The abstract was improved by the following:

  • we replaced the sentence: “The influence of Co2+ ions substitution with Ni2+ ions on the structural, morphological and optical properties of Co(1-x)NixCr2O4 (x=0.25, 0.5, 0.75 and 1) was investigated. ” with ”The structural, morphological and electronic properties of Ni2+ ions in CoCr2O4 matrix as ceramic pigments were investigated.”

  • we replaced the sentence: “XRD studies reveal a spinel type structure for all Co(1-x)NixCr2O4 samples and a secondary Cr2O3 phase was found in all the Co2+ and Ni2+ samples (x≤0.75).” with ”The X-ray diffraction studies (XRD) reveal a spinel-type face-centered cubic structure and a secondary Cr2O3 phase when x≤0.75 and a body-centered tetragonal structure when x=1.”

  • the following phrases were added to the abstract at the reviewer suggestion: “Fourier Transform Infrared Spectroscopy (FTIR) reveled two strong absorption bands corresponding to the metal-oxygen stretching from tetrahedral and octahedral sites, characteristic of spinel structure. Ultraviolet-Visible (UV–Vis) spectra exhibited the electronic transitions of the Cr2+ Cr3+ and Ni2+ ions.”

  •  

4. The introduction is not clear enough and it is difficult for the reader to find a relation between ideas with the text in general. Some parts of the current introduction lack consistency and are not easy to follow and should be rewritten. Furthermore, the authors should emphasize their novelties concerning others in this field. They must underline what is the motivation to start to study exactly this topic. Besides they should also mention explicitly, what the purpose of this work is and what they aim at. Besides, it does not provide an appropriate context for reviewing the literature and does not properly answer the question 'Why'.

As the reviewer’s suggestion the introduction was improved upon in terms of consistency. The aim and the novelty of the present work were also described in a clear manner.

5. In the manuscript, please avoid using "etc."

The authors removed “etc.” from 2 of the sentences and restructured the phrase “Spinels have the general formula AB2O4 (A=Co, Ni, Mn, Zn and not only), where the A-sites are tetrahedral coordinated occupied by divalent ions as A2+, and the B-sites are octahedral coordinated and occupied by Cr3+ ions [16]”. with: ”Spinels have the general formula AB2O4 where the A-sites are tetrahedral coordinated occupied by divalent ions as A2+ (like Co, Ni, Mn, Zn), and the B-sites are octahedral coordinated and occupied by trivalent ions like Cr3+ [16].”

6. In the experimental section please give more explanation about how the synthesized samples have been measured for their optical properties.

At the reviewer suggestion, the authors added to the manuscript the following text in the experimental section: “The absorption spectra of the powder samples were obtained using a Perkin-Elmer Lambda 45 UV/Vis spectrometer with integrating sphere using the pellet technique. The powder samples mixed with BaSO4 were uniaxial press in a pellet matrix using a load force of 10 tons/cm2 to form transparent disks with diameter of 13mm, and 2mm thick. The spectra were recorded in the 200–850 nm wavelength range, with the wavelength accuracy ±2 nm.”

And the following sentence was removed: “The optical absorption spectra of the samples were obtained using a Perkin-Elmer Lambda 45 UV/Vis spectrometer with integrating sphere. The spectra were recorded in the 200–850 nm wavelength range, and a ±2 nm validity of the band position.”

7. The manuscript contains relatively a high proportion of self-citations.

The authors restructured the bibliography and removed 3 from 5 auto-citations in the first manuscript. The remaining auto citations are relevant for the present work and cannot be excluded.

8. Please provide a larger view of the FTIR spectra in the range of 500-700 cm-1 for all samples for comparison.

The Figure 5. - FT- IR spectra of Co(1-x)NixCr2O4 was improved adding also the spectra of CoCr2O4 (x=0, 0.25, 0.5, 0.75 and 1) and an insertion was added which unfold from 500 cm-1 to 700 cm -1 providing a larger view, as reviewer suggested.

9. I recommend the authors discuss deeply the obtained results by linking them together to get a scientific finding.

The manuscript was improved.

10. Rietveld analysis plots must be "Figure 3" instead of "Figure 2".

The Figure entitled “Rietveld analysis of powder-diffraction pattern” was corrected numbered to Figure 3.

11. In Figure 2. (especially XRD patterns for NiCr2O4), some main and intense peaks seem to be split into two peaks. please explain about it.

The authors acknowledge the reviewer’s remarks and provided some additional information to the XRD section: “The Co(1-x)NixCr2O4 spinel (when x ≠1) crystallizes in Face-Centered Cubic structure and is indexable to the space group Fd-3m (no. 227). This is consistent with the standard values of face-centered cubic phase (00-022-1084 PDF files). On the other hand, NiCr2O4 crystalizes in the Body-Centered Tetragonal structure, the space group I41/amd (no. 141). So, a transition from the cubic spinel (when x<1) to a body centered tetragonal structure (when x=1) is observed.” which explain the peaks from the NiCr2O4 diffraction pattern.

Also the following sentence was added: The XRD peaks at 2θ values observed in the NiCr2O4, pattern, match the (112), (200), (103), (211), (202), (004), (220), (312), (106), (321), (224) and (400) crystalline places of the body-centered tetragonal spinel structure (01-088-0109 PDF files).”, to clarify the NiCr2O4 diffraction pattern.

12. The quality of Figure 4 and Table 1 are not acceptable and suitable. Table 1 is not well organized.

We restructured Table 1 in two distinct tables to facilitate the results interpretation. Table 1 includes the average crystallite size and cell parameters, and Table 2 includes information regarding the mass and density phase. Also, the resolution of Figure 4 was improved.

13. In Figures 2 and 3 the authors talk about 5 samples despite other Figures. It's kind of confusing for readers. Moreover, in the whole manuscript some sentences are vague, please express them vividly and adjust all of them to a unique state (from the point of view of the number of samples). For instance:

"varying the quantity of Ni2+ ions in the CoCr2O4 spinel matrix"

"XRD patterns of different Co2+ substitution with Ni2+ ions in CoCr2O4 spinel matrix"

"The influence of Co2+ ions substitution with Ni2+ ions on the structural, morphological and optical properties of Co(1-x)NixCr2O4 (x=0.25, 0.5, 0.75 and 1) was investigated"

The manuscript was improved with adding additional results FTIR, UV-Vis, SEM and TEM for the CoCr2O4 pigment powder.

To avoid confusion, the syntax “Co2+ ions” was eliminated from the manuscript and we remain consistent with the syntax “Ni2+ ion substitution”.

14. The conclusions of the work don’t convince me and it should be rewritten. The conclusion must be quantitative and comparative. The authors should clarify their findings in this section. Please state the most valuable sample among the other samples that gave the highest/good performance. Besides, the conclusion must be in the past tense.

The authors have reformulated the conclusions to be more clearer and more quantitative as per reviewer’s suggestion.

15. Finally, there are some grammatical, typos, and technical errors. besides a few unclear sentence structures that exist in the whole manuscript. Please proofread them before more consideration.

The authors improved all the manuscript paying a special attention to the English language.

Respectfully, we thank the Editor for his thoughtful cooperation, and we believe our R1 revised manuscript can meet the journal’s high standards and will be acceptable for publication.

On behalf of authors,

Raluca Anca Mereu

Reviewer 3 Report

Goga et al. described the effect of the substitution of Co2+ ions by Ni2+ ions on the structural, morphological and optical properties of Co(1-x)NixCr2O4 (x=0.25, 0.5, 0.75 and 1). The samples were synthesized using sol-gel technique. Their thermal, structural, and morphological features were investigated and their stability, coloring properties, and potential applications in ceramics were tested. The manuscript can be published in the journal after major revision. Please, see my comments below.

1. I advise the authors to carefully revise the language and style throughout the manuscript, because there are hard to read and incomprehensible sentences, repetitions of sentences in the text. There are two “Figure 2” in the text.

2. Data on DTA is uncertain. Please, provide Tonset, Tpeak, Tend.

3. Mark the Cr2O3 peaks in the XRD patterns containing cobalt.

4. If NiCr2O4 and CoCr2O4  are isostructural their XRD patterns should be the same, however the NiCr2O4 pattern shows additional peaks compared to the CoCr2O4 one (the authors marked them on the diffractogram). Please, explain.

Also, experimental curve for NiCr2O4 curve does not fit well with the structural model (Fig. 2e). What were the R factors? Please, provide cell parameters for NiCr2O4.

5. The authors wrote that “peak centered at 516 cm-1 from sample with x=0.25 shifts to lower frequency, at 504 cm-1, with the increasing nickel content”, while “the second peak at 620 cm-1 shifts towards higher frequencies with the increase of nickel content”. According to Figure 5, both peaks are shifted to one side, but the shift of the second peak is greater.

6. Since the CoCr2O4 sample has been synthesized, I suggest providing its SEM and TEM images for comparison with other samples as well as UV-VIS spectra.

7. It is desirable to highlight what fundamentally new results were obtained by the authors in comparison with the literature data.

Author Response

Respectfully, authors thank reviewer no. 3 for the evaluation of our research work, the appreciation of our research results and recommendation for publication in Materials after major revision.

Please see below and attached the authors comments:

1. I advise the authors to carefully revise the language and style throughout the manuscript, because there are hard to read and incomprehensible sentences, repetitions of sentences in the text. There are two “Figure 2” in the text.

The authors improved all the manuscript paying a special attention to the English language.

2. Data on DTA is uncertain. Please, provide Tonset, Tpeak, Tend.

At the suggestion of the reviewer we included the following sentence in the Differential thermal analysis section: “The peak corresponding to the elimination of the organic fragments can be described by the following temperatures: Tonset=318.35 °C, Tpeak =331.02 °C and Tend= 352.76 °C.”

3. Mark the Cr2O3 peaks in the XRD patterns containing cobalt.

As the reviewer suggested, the Cr2O3 peaks were evidence in Figure 2.

4. If NiCr2O4 and CoCr2O4 are isostructural their XRD patterns should be the same, however the NiCr2O4 pattern shows additional peaks compared to the CoCr2O4 one (the authors marked them on the diffractogram). Please, explain. Also, experimental curve for NiCr2O4 curve does not fit well with the structural model (Fig. 2e). What were the R factors? Please, provide cell parameters for NiCr2O4.

The authors acknowledge the reviewer’s remarks and provided some additional information to the XRD section: “The Co(1-x)NixCr2O4 spinel (when x ≠1) crystallizes in Face-Centered Cubic structure and is indexable to the space group Fd-3m (no. 227). This is consistent with the standard values of face-centered cubic phase (00-022-1084 PDF files). On the other hand, NiCr2O4 crystallizes in the Body-Centered Tetragonal structure, the space group I41/amd (no. 141). So, a transition from the cubic spinel (when x<1) to a body centered tetragonal structure (when x=1) is observed.” which explain the peaks from the NiCr2O4 diffraction pattern.

Also the following sentence was added: The XRD peaks at 2θ values observed in the NiCr2O4, pattern, match the (112), (200), (103), (211), (202), (004), (220), (312), (106), (321), (224) and (400) crystalline planes of the body-centered tetragonal spinel structure (01-088-0109 PDF files).”, to clarify the NiCr2O4 diffraction pattern.

R-factors (not corrected for background) for Pattern:  1

=> Rp: 29.5 Rwp: 37.8 Rexp: 29.31 Chi2: 1.66 L.S. refinement

=> Conventional Rietveld R-factors for Pattern: 1

=> Rp: 38.0 Rwp: 45.8 Rexp: 35.51 Chi2: 1.66

We restructured Table 1 in two distinct tables to facilitate the results interpretation. Table 1 includes the average crystallite size and cell parameters, and Table 2 includes information regarding the mass and density phase. Thereby the cell parameters for NiCr2O4 are listed in Table 1.

5. The authors wrote that “peak centered at 516 cm-1 from sample with x=0.25 shifts to lower frequency, at 504 cm-1, with the increasing nickel content”, while “the second peak at 620 cm-1 shifts towards higher frequencies with the increase of nickel content”. According to Figure 5, both peaks are shifted to one side, but the shift of the second peak is greater.

The authors corrected the interpretation of the data regarding Figure 5.

6. Since the CoCr2O4 sample has been synthesized, I suggest providing its SEM and TEM images for comparison with other samples as well as UV-VIS spectra.

At the suggestion of the reviewer the authors included the UV-Vis spectra, SEM and TEM images.

7. It is desirable to highlight what fundamentally new results were obtained by the authors in comparison with the literature data.

The authors highlighted the novelty of the present study when compared with literature.

Respectfully, we thank the Editor for his thoughtful cooperation, and we believe our R1 manuscript can meet the journal’s high standards and will be acceptable for publication.

On behalf of authors,

Raluca Anca Mereu

Round 2

Reviewer 2 Report

After the authors' careful revision the manuscript seems fluent now. The authors have responded properly to the comments and suggestions. No further amendment is needed. The present form of the manuscript can be considered for publication.

Reviewer 3 Report

Authors answered most of my comments. The paper can be accepted for publication.